# Determination of the Prevalence and Antimicrobial Resistance of *Enterococcus faecalis* and *Enterococcus faecium* Associated with Poultry in Four Districts in Zambia

**DOI:** 10.3390/antibiotics12040657

**Published:** 2023-03-28

**Authors:** Grace Mwikuma, Henson Kainga, Simegnew Adugna Kallu, Chie Nakajima, Yasuhiko Suzuki, Bernard Mudenda Hang’ombe

**Affiliations:** 1Department of Pathology, Kitwe Teaching Hospital, Kitwe 10101, Zambia; 2Department of Paraclinical Studies, School of Veterinary Medicine, University of Zambia, Lusaka 10101, Zambia; 3Department of Veterinary Epidemiology and Public Health, Faculty of Veterinary Medicine, Lilongwe University of Agriculture and Natural Resources, Lilongwe 207203, Malawi; 4College of Veterinary Medicine, Haramaya University, Dire Dawa P.O. Box 138, Ethiopia; 5Division of Bioresources, International Institute for Zoonosis Control, Hokkaido University, Sapporo 060-0808, Japan; 6International Collaboration Unit, International Institute for Zoonosis Control, Hokkaido 19 University, Sapporo 060-0808, Japan; 7Hokkaido University Institute for Vaccine Research and Development, Hokkaido 19 University, Sapporo 060-0808, Japan

**Keywords:** antimicrobial resistance, antimicrobial resistance genes, *Enterococcus faecalis*, *Enterococcus faecium*, prevalence, poultry, Zambia

## Abstract

The presence of antimicrobial-resistant *Enterococci* in poultry is a growing public health concern worldwide due to its potential for transmission to humans. The aim of this study was to determine the prevalence and patterns of antimicrobial resistance and to detect drug-resistant genes in *Enterococcus faecalis* and *E. faecium* in poultry from four districts in Zambia. Identification of *Enterococci* was conducted using phenotypic methods. Antimicrobial resistance was determined using the disc diffusion method and antimicrobial resistance genes were detected using polymerase chain reaction and gene-specific primers. The overall prevalence of *Enterococci* was 31.1% (153/492, 95% CI: 27.1–35.4). *Enterococcus faecalis* had a significantly higher prevalence at 37.9% (58/153, 95% CI: 30.3–46.1) compared with *E. faecium*, which had a prevalence of 10.5% (16/153, 95% CI: 6.3–16.7). Most of the *E. faecalis* and *E. faecium* isolates were resistant to tetracycline (66/74, 89.2%) and ampicillin and erythromycin (51/74, 68.9%). The majority of isolates were susceptible to vancomycin (72/74, 97.3%). The results show that poultry are a potential source of multidrug-resistant *E. faecalis* and *E. faecium* strains, which can be transmitted to humans. Resistance genes in the *Enterococcus* species can also be transmitted to pathogenic bacteria if they colonize the same poultry, thus threatening the safety of poultry production, leading to significant public health concerns.

## 1. Introduction

*Enterococcus* is a genus of Gram-positive bacteria in the family Enterococcaceae, the order Lactobacillales and the phylum Firmicutes [1]. *Enterococcus* is part of the normal flora in the gastrointestinal tract (GIT) of mammals, fish, reptiles, insects, and birds [2,3]. Being ubiquitous in nature, it is also found in soil, plants, sewage and fresh and salt water [4,5]. Species in the genus *Enterococcus* (E) have emerged as pathogens of medical and public health importance [6]. This is partly due to their adaptability to the selective pressures of antimicrobials. They also have the ability to acquire, express, and transmit mobile genetic elements (MGEs) from/to pathogenic as well as non-pathogenic species in the same or different genus [7,8], leading to the development of antimicrobial resistance. MGEs play an important role in facilitating horizontal genetic exchange and promoting the acquisition and transmission of resistance genes [9]. These properties have made *Enterococcus* an important human pathogen responsible for a number of clinical conditions, including urinary tract infections (UTI), endocarditis, bacteremia and mastitis in humans and animals [10,11]. *Enterococcus* species also cause locomotive disorders and septicemia in broilers [12]. *Enterococci* is ranked among the major causes of nosocomial infections worldwide [13]. This is especially true for *Enterococcus (E) faecalis* and *E. faecium*. The emergence of multidrug-resistant (MDR) *Enterococci* such as vancomycin-resistant *Enterococci* (VRE) and drug-resistant *Enterococci* in poultry are of major public health concern because of the limited treatment options available for infections caused by such species, as well as the possibility of dispersion between poultry and humans [3,14,15,16] and the transfer of resistance genes to other bacteria (9). This has led to an increase in infections caused by multidrug-resistant *Enterococci*, which can not only be very difficult to treat but can also lead to increased mortality rates [17].

Enterococcal infections can be serious and are associated with increased healthcare costs, including the cost of hospitalization, laboratory testing and antibiotic treatment [18]. Enterococcal infections can also lead to lost productivity due to missed work or school.

Although *Enterococcus faecalis* and *Enterococcus faecium* are commonly found in the guts of poultry, they can cause infections in poultry that can lead to significant economic losses for the industry. Enterococcal infections in poultry can result in decreased growth rates, reduced feed efficiency and increased mortality rates [19]. Poultry and food products of poultry origin are the most consumed worldwide [20]. *Enterococci* can contaminate poultry products and pose a risk to human health if consumed [21]. Antibiotic resistance in *Enterococci* is also a concern for the poultry industry, as the use of antibiotics in poultry production can contribute to the development and spread of antibiotic-resistant strains [22]. Therefore, the presence of antimicrobial-resistant *Enterococci*, especially multidrug resistance *Enterococcus* species, in poultry is of public health concern as it may serve as a pool from which antimicrobial resistance genes are disseminated. VRE is a nosocomial pathogen that exhibits multidrug resistance (MDR) and virulence. 

*Enterococcus faecium* has transitioned from a commensal organism to an ESKAPE (*E. faecium*, *Staphylococcus aureus*, *Klebsiella pneumoniae*, *Acinetobacter baumannii*, *Pseudomonas aeruginosa*, and *Enterobacter species*) pathogen. ESKAPE is an acronym for a group of life-threatening nosocomial pathogens that successfully evade the effect of antimicrobial drugs and represent a model for pathogenesis, transmission, and resistance [23]. VRE cause a greater number of infections than other nosocomial pathogens in hospitals in the United States [23].

Vancomycin-resistant *Enterococci* have been reported worldwide [24], including in Zambia [25]. However, they have not been given the same attention as other commensals of the GIT such as *Staphylococci*, *Salmonella*, *Shigella*, *Campylobacter* and *Escherichia coli*. Zambia developed a multi-sectoral national action plan in recognition of the public health threat of morbidity, mortality, and economic outcomes of antimicrobial resistance. However, minimal surveillance and research have been conducted on MDR *Enterococci* in Zambia. This study aimed to determine the prevalence of antimicrobial resistance and the presence of antimicrobial-resistant genes in *Enterococcus faecalis* and *Enterococcus faecium* isolates from poultry in four districts in Zambia.

## 2. Results

### 2.1. Identification

#### 2.1.1. Identification of *Enterococci* Using Analytical Profile Index (API)

Of the 37 poultry isolates subjected to API identification using BioMérieux’s Analytical Profile Index (API) 20 Strep test kits, 19 were identified as *Enterococcus faecalis*, 15 as *Enterococcus faecium* and one as *Enterococcus durans*. Two were not identified. The reason for performing API tests on only 37 isolates was due to insufficient reagents. Particularly, the NIN, VP 1 + VP 2, ZYM A and ZYM B were enough for only 38 samples (one control *Enterococcus faecalis* ATCC 29212 strain and the 37 isolates).

#### 2.1.2. Identification of *Enterococci* Using Polymerase Chain Reaction (PCR)

PCR was run on 343 suspect *Enterococcus* DNA samples extracted from poultry droppings using genus-specific primers for elongation factor (*tuf*) and d-alanine-d-alanine ligase (*ddl*) genes. PCR was subsequently run on 153 positive DNA samples using species-specific primers for *Enterococcus faecalis* and *Enterococcus faecium*. The most common *Enterococcus* species was *E. faecalis* (37.9%), followed by *E. faecium* (10.5%). Remarkably, 38.6% of the isolates contained more than one species, with 34.6% of the total enterococcus isolates containing both *E. faecalis* and *E. faecium*. Adding the latter to *E. faecalis* and *E. faecium*, *E. faecalis* would still be the most predominant species, followed by *E. faecium* (Figure 1). The word “Other” represents *Enterococcus* species—identified by PCR using genus-specific *ddl* and *tuf* gene primers—which could not be identified through PCR due to lack of additional species-specific primers, or DNA sequencing due to the unavailability of reagents. Figure 1 shows the species identified using *E. faecalis* and *E. faecium* species-specific primers. 

#### 2.1.3. Comparing API and PCR Identification

API and PCR results were compared to ascertain the agreement between the two methods. API correctly identified 17 (45.9%) of the 37 isolates. API could not identify isolates with more than one species and only picked one of the species in samples with two or more species (16, 43.2%). It also misidentified an isolate that contained *E. faecalis* and another species as *E. faecium*, and it was not able to identify two isolates. Additionally, API identified one isolate as *E. durans1*, but this could not be confirmed as the corresponding species-specific primers were not available (Table 1). 

### 2.2. Prevalence of Enterococci

#### 2.2.1. Overall Prevalence

The overall prevalence of *Enterococci* was 31.1% (153/492, CI: 27.1–35.4), while the prevalence in Lusaka Province was 30.8% (33/107, CI: 22.5–40.6) and the prevalence in Copperbelt Province was 31.2% (120/385, 26.6–36.1). Table 2 contains summaries of the prevalence of *E. faecalis* and *E. faecium* (combined and separate) in poultry from districts in the Copperbelt and Lusaka Provinces.

#### 2.2.2. Species-Specific Prevalence of Isolates

The prevalence of *Enterococci* varied significantly across the districts. Lusaka district had the highest prevalence at 44.0% (22/50, CI: 30.3–58.7) compared with the other three districts of Kitwe, Ndola and Chongwe (*p* = 0.038). The prevalence of *E. faecalis* was higher than that of *E. faecium* in all districts (*p* = 0.012) except Kitwe district (*p* = 0.044) (Table 3).

### 2.3. Antimicrobial Susceptibility Test Results

#### 2.3.1. Antimicrobial Susceptibility of *Enterococci*

All intermediate test results were considered resistant. Both *Enterococcus* species showed very high (97.3%) resistance to tetracycline, while 94.6% were resistant to erythromycin and 77.0% were resistant to ciprofloxacin. Remarkably, 64.9% of both *Enterococcus* species were susceptible to vancomycin. More than 90.0% of *E. faecalis* isolates were resistant to erythromycin and tetracycline and more than 50.0% were resistant to ampicillin, chloramphenicol and ciprofloxacin. Less than 20.0% of the *E. faecalis* isolates were resistant to vancomycin. All *E. faecium* isolates in this study were resistant to erythromycin. More than 80.0% of *E. faecium* isolates exhibited phenotypic resistance to ampicillin, ciprofloxacin and tetracycline, while less than 40.0% showed resistance to chloramphenicol and vancomycin. Susceptibility profiles of *E. faecalis* and *E. faecium* to the eight antimicrobials tested are provided in Table 4. 

#### 2.3.2. Number of Enterococcus Isolates Resistant to One, Two, Three or More Antimicrobial Classes

Multidrug resistance (MDR) is defined as resistance to three or more classes of antimicrobials. The results of our study show that none of the isolates were susceptible to all antimicrobial classes tested. Of the 74 *E. faecalis* and *E. faecium* isolates tested against eight antimicrobials, only two (2.7%) were resistant to one class of antimicrobials. A total of 5 (6.8%) isolates were resistant to two classes of antimicrobials. The majority of *E. faecalis* and *E. faecium* isolates (67, 90.5%) were MDR (Table 5). 

### 2.4. Detected Antimicrobial Resistance Genes

#### 2.4.1. Presence of Antimicrobial Resistance Genes in *E. faecalis* and *E. faecium*

The *aac(6′)-Ie-aph(2″)-LA* resistance gene encoding resistance to gentamycin was detected in 33 and 12 *E. faecalis* and *E. faecium* isolates, respectively, representing 60.8% of the isolates. The *ermB* resistance gene was more common in both *E. faecalis* and *E. faecium* compared with the *ermA* gene. The *vanA* resistance gene was detected in only two *E. faecalis* isolates and in none of the *E. faecium* isolates. Table 6 shows the number of different resistance genes detected in the *E. faecalis* and *E. faecium* isolates.

#### 2.4.2. Resistance Genes in *E. faecalis* and *E. faecium* isolates across the Study Area

The most common resistance genes in *E. faecalis* isolates from Chongwe district in Lusaka Province were *tetL* and *tetM*, as they were found in all five isolates. These were followed by *aac* and *tetK*, which were detected in four of the five isolates, and *ermB*, which was detected in three isolates. The most common resistance genes in isolates from Lusaka district were *tetK* and *tetM*, which were found in all 8 *E. faecalis* isolates. These were followed by *tetL* (7/8), *aac* (6/8) and *ermB* (5/8). Resistant genes commonly detected in *E. Faecalis* from Ndola district were *tetM* (16/20), *tetL* (15/20), *aac* (13/20), *ermB* (12/20) and *tetK* (11/20). In *E. faecalis* isolates from Kitwe, *tetK* (17/25) was the most prevalent resistance gene, followed by *tetM* (16/25), *ermB* (15/25), *tetL* (14/25) and *aac* (10/25). The most commonly detected resistance genes in *E. faecium* isolates from Kitwe district were *aac* (5/8) and *ermB* (5/8), followed by *tetM* (4/8) and *tetK* (3/8). In Lusaka district, the most common genes were *aac* (5/5) followed by *ermB* (4/5), *tetL* (4/5), *tetK* (3/5) and *tetM* (3/5). Table 7 shows all resistance genes detected in *E. faecalis* and *E. faecium* isolates from the four districts in Zambia.

### 2.5. Association between Antimicrobials and Resistance Genes

Differences in antimicrobial resistance patterns and resistance genes in both enterococcus species were analyzed to assess possible associations between resistance phenotypes and their corresponding genotypes. A positive association between phenotype and genotype was found for tetracycline (*p* = 0.047) and erythromycin (*p* = 0.008), but there was no association between genotype and the vancomycin resistance phenotype (*p* = 0.051) (Table 8).

## 3. Discussion

The prevalence, antimicrobial susceptibility patterns and presence of resistance genes in poultry droppings from four districts in Zambia were determined. The overall prevalence of *Enterococci* was 31.1%. This is in agreement with other studies that reported similar results in Poland [26], Malaysia [27] and Nigeria [28]. However it was lower than that reported in a similar study conducted in Zambia, where the prevalence was 88.4% in laying hens [29]. This could be due to differences in sampling methods, farms sampled and the number of farms sampled. Another previous study [30] also reported a higher prevalence than that reported in the present study. Conversely, the prevalence rate in our study was higher than the rates reported in Ethiopia [31], Pakistan [32] and Thailand [33]. The differences in the isolation rates of *Enterococci* can be attributed to several factors, including antibiotic use, environmental factors and methodology. The widespread use of antibiotics has led to the selection and dissemination of antibiotic-resistant *Enterococci*. *Enterococci* are found in soil and water and can persist in the environment for long periods of time, making them more difficult to control and leading to increased isolation rates. The isolation rate can also be influenced by the type of culture method used and the presence of selective media that may preferentially isolate *Enterococci* [34].

Among the *Enterococcus* species isolated in this study, *E. faecalis* was the most prevalent (37.9%), followed by *E. faecium* (10.5%). This was in agreement with other studies [35,36,37,38] which found *E. faecalis* to be the most prevalent species in poultry. However, our study differed slightly from some studies that found species other than *E. faecalis* to be the most predominant [16,39,40]. The variations in species levels between studies might be due to differences in the type of poultry, source of chicks, sampling methods, geographical disparities, the time of study and isolation and identification procedures [40].

Although API 20 strep is considered the best identification system for bacteria [41], it does not accurately identify some species of *Enterococci* [42]. In the present study, we validated API 20 strep results using PCR. PCR conducted using species-specific primers identified 91.9% of samples containing both single and multiple species. API 20 strep accurately identified 45.3% of *Enterococcus* species, but identified only one species in isolates containing more than one species. It also misidentified 2.7% of the *Enterococcus* species. Our findings were in agreement with the results of previous studies [42,43,44,45]. 

In the present study, phenotypic resistance to critically important antimicrobials, as defined by the WHO [46], was observed and 90.5% of *E. faecalis* and *E. faecium* isolates were multidrug resistant (MDR) (Table 4). Notably, all *E. faecalis* and *E. faecium* isolates were resistant to one or more of the tested antimicrobials (Table 5). These findings were similar to those of a study done earlier [47] in which the majority of *E. faecalis* and *E. faecium* isolates were resistant to one or more of the tested antimicrobials. Resistance to all tested antimicrobials was also observed in both *E. faecalis* and *E. faecium* isolates.

More than 50.0% of *E. faecalis* isolates were resistant to all tested antimicrobials, while 100.0% of the isolates were resistant to tetracycline. On average, *E. faecium* exhibited increased resistance to antimicrobials in comparison with *E. faecalis*. Our findings are in agreement with a recent study conducted in Zambia [29]. Our study also has some similarities with a study conducted in the Czech Republic [48], in which increased resistance of *Enterococci* to tetracycline, erythromycin and nitrofurantoin were observed, as well as a study from USA [49], in which *Enterococci* resistance to tetracycline, penicillin and ciprofloxacin was documented. Furthermore, our results are consistent with findings from previous studies [50,51,52,53,54,55,56] where high tetracycline resistance was reported. Our results were also comparable with those of the study by Fracalanzza et al. [57], which recorded the resistance of *Enterococci* to erythromycin to be at 82.0% when intermediate results were included. Nevertheless, that study noted reduced resistance to tetracycline (38.3%) and chloramphenicol (5.7%) compared with our study. The observed increase in resistance to all the antimicrobials tested indicate that poultry from these four districts in Zambia can be a source of MDR *Enterococci*. However, our study contrasted with other studies [58,59], which reported lower levels of resistance to antimicrobials.

Although the gene *aac(6′)-Ie-aph(2″)-LA*, which encodes resistance to gentamicin, was detected in 60.8% of both *Enterococci* species tested, an association with susceptibility could not be determined, as discs containing high concentrations of gentamicin (for example 120 μg or 500 μg), which are used to detect high-level aminoglycoside resistance, were not available. 

The associations between antimicrobial resistance phenotypes and genotypes in *E. faecalis* and *E. faecium* isolates were analyzed. Associations were found between genotypes and tetracycline and erythromycin resistant phenotypes. However, genotypes showed no relationship with vancomycin resistant phenotypes. The disparity observed between the phenotypes and genotypes in the case of vancomycin could be due to the fact that vancomycin resistance in *Enterococci* can be conferred by different gene clusters [60,61,62]. 

## 4. Materials and Methods

### 4.1. Study Design and Sites

A cross-sectional study was conducted in selected farms in Chongwe and Lusaka (Lusaka Province) and Ndola and Kitwe (Copperbelt Province) districts in Zambia (Figure 2). The two provinces are among those that harbor most of the commercial poultry farms in Zambia.

### 4.2. Sample Collection

A total of 492 freshly voided poultry droppings were collected from layers, broilers and village chickens. Five different visits were made to selected poultry farms in four districts in the Copperbelt and Lusaka Provinces in Zambia (Figure 2). Of the total samples collected, 57 were from farms in Chongwe, while 50 were from Lusaka district in Lusaka Province. Of the 385 samples from Copperbelt Province, 140 and 245 came from Ndola and Kitwe districts, respectively. 

### 4.3. Laboratory Investigations

#### 4.3.1. Isolation of *Enterococci*

Conventional microbiological assays were performed to detect and identify *Enterococcus* species as described by Facklam and Collins [63]. Briefly, 1 g of poultry droppings was suspended in 9 mL buffered peptone water (BPW) (HIMEDIA, India), mixed and incubated at 37 °C for 24 h. A loopful of the BPW suspension was streaked on Bile Esculin Agar (BEA) (HIMEDIA, India) and incubated at 37 °C for 24 h. Following this, colonial traits were noted and smears of suspect colonies (small black shiny colonies on BEA) were made and stained using Central Drug House’s (CDH) Gram’s color staining kit from India. Gram-positive cocci appearing in chains, doubles or singles were characteristic of enterococci. A total of 343 suspected *Enterococcus* isolates were recovered from the 492 samples tested. These were stored in 20% glycerol at −20 °C for subsequent experiments.

#### 4.3.2. Identification of *Enterococci* Using Analytical Profile Index (API)

Species identification based on phenotypic characteristics and biochemical tests was conducted using BioMérieux’s Analytical Profile Index (API) 20 Strep test kits. A total of 37 isolates were identified using the API 20 Strep test kits. The reasons for this are stated in Section 2.1.1.

#### 4.3.3. DNA Extraction

Colonies grown overnight on a blood agar plate were placed in a test tube containing 0.5 mL of molecular grade water, vortexed and boiled at 95 °C for 10 min and then centrifuged for 5 min at 1500× *g*. The supernatant was pipetted into cryo-vials and stored at −20 °C for further analysis.

#### 4.3.4. Molecular Identification of *Enterococci*

Molecular identification of the *Enterococcus* species was conducted using single PCR and the genus-specific and species-specific primers shown in Table 9, following the procedure described by Li et al. (2012) [64]. PCR amplification of *elongation factor* (*tuf*) and *D -Ala- D -Ala ligase* (*ddl*) in the extracted DNA was conducted using Phusion Flash High-Fidelity PCR Master Mix (Thermofisher Scientific, USA) in a thermal cycler (Applied Biosystems, Chiba, Japan) under the following PCR conditions: initial denaturation at 98 °C for 2 min followed by 30 cycles of denaturation at 98 °C for 5 s, annealing at 56 °C for 5 s, and extension at 72 °C for 30 s. A final extension was performed at 72 °C for 1 min. PCR amplicons were run on 1.5% agarose gels. The expected bandwidths for *tuf* and *ddl* PCR products were 112 bp and 475 bp, respectively. For species identification, species-specific primers (Table 1) targeting the superoxide dismutase (*sodA*) gene in *E. faecalis* and *E. faecium* were used. No primers were available for other species. The PCR conditions were similar to those used for genus amplification, except for the annealing temperature, which was set to 52 °C for both species. 

#### 4.3.5. Determination of Antimicrobial Resistance Levels

Susceptibility to vancomycin (30 μg), erythromycin (15 μg), ampicillin (10 μg), penicillin (10 U), tetracycline (30 μg), nitrofurantoin (300 μg), ciprofloxacin (5 μg) and chloramphenicol (30 μg) was determined using the disk diffusion method according to the Clinical and Laboratory Standards Institute guidelines [68]. The disks used for susceptibility testing were manufactured by HIMEDIA, India. Diameters of the zones of inhibition were recorded in millimeters (mm) and interpreted as susceptible, intermediate or resistant. In this study, intermediate results were taken as resistant. A reference strain, *Enterococcus faecalis* 29,212, was used as a control strain.

#### 4.3.6. Detection of Antimicrobial Resistant Genes (ARG)

Genes conferring resistance to aminoglycosides [*aac(6′)-le-aph(2″)-LA*], which in this study was abbreviated as “*aac*”, macrolides (*ermA* and *ermB*), tetracyclines (*tetM*, *tetL*, *tetK*, and *tetX*) and glycopeptides (*vanA*) were detected in a single PCR using the gene-specific primers shown in Table 10. One Taq Quick-load 2X Master Mix (Biolabs, Durham, NC, USA) was used for amplification in a thermal cycler (Applied Biosystems, Chiba, Japan). The following PCR conditions were employed: initial denaturation at 93 °C for 3 min followed by 35 cycles of denaturation at 93 °C for 60 s, annealing at 52 °C for 60 s and elongation at 72 °C for 60 s. The final elongation step was performed at 72 °C for 5 min. PCR amplicons were run on 1.5% agarose gels. The expected sizes of the PCR products differed for each gene (Table 10). 

### 4.4. Data Analysis

Data were entered, cleaned and validated in a Microsoft™ excel spreadsheet (MS Office Excel^®^ 2016). The data were then exported to SPSS software ver. 21 (IBM Corp., Armonk, NY, USA). PCR results (positive or negative) were reference variables for descriptive analyses. Univariate analyses were conducted for descriptive statistics and data are presented as frequencies, percentages and prevalence. 

## Figures and Tables

**Figure 1 antibiotics-12-00657-f001:**
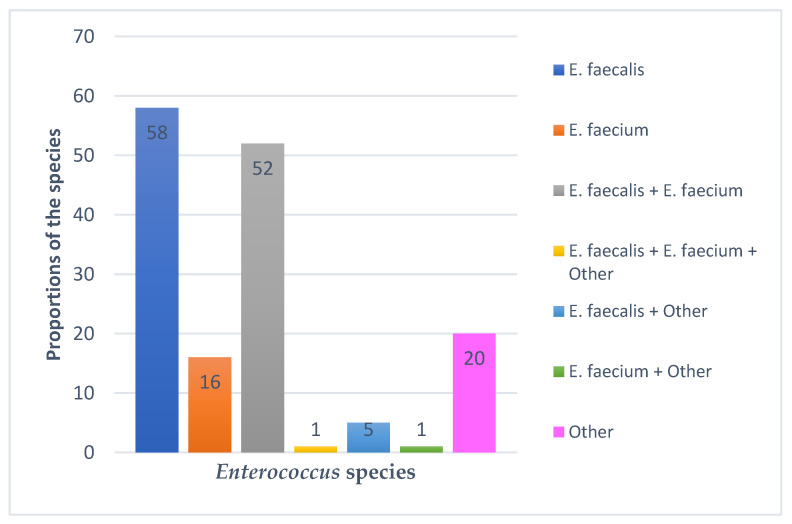
*Enterococcus* species identified using species-specific *E. faecalis* and *E. faecium* primers.

**Figure 2 antibiotics-12-00657-f002:**
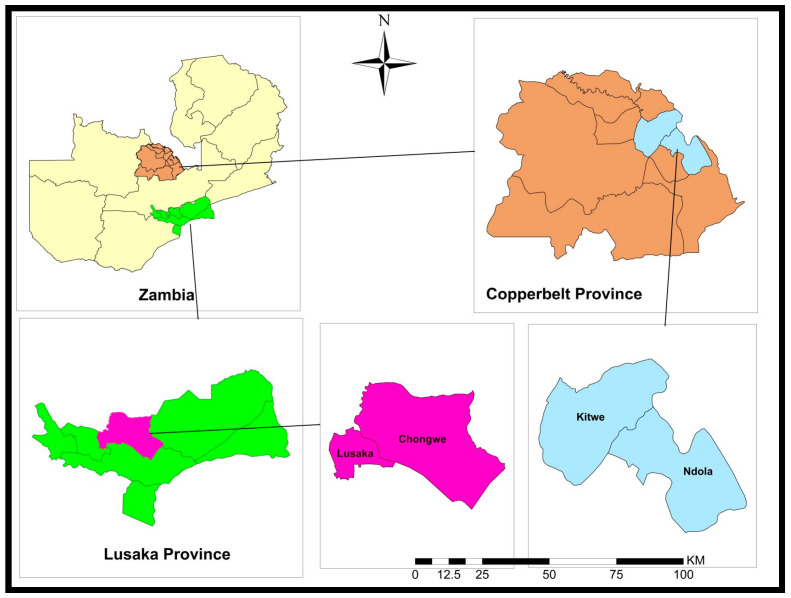
Map of the study areas.

**Table 1 antibiotics-12-00657-t001:** API and PCR Identities.

Study ID	PCR ID	API ID	Study ID	PCR ID	API ID
80	*E. faecalis*	*E. faecalis*	454	*E. faecalis* + *E. faecium*	*E. faecium*
82	*E. faecium*	*E. faecium*	455	*E. faecalis* + *E. faecium*	*E. faecalis*
84	*E. faecalis* + *E. faecium*	*E. faecalis*	476	*E. faecalis* + *E. faecium*	*E. faecium*
87	*E. faecalis*	*E. faecalis*	477	*E. faecalis* + *E. faecium*	*E. faecalis*
88	*E. faecalis* + Other	*E. faecium*	501	Other enterococci	Not identified
89	Other enterococci	Not identified	552	*E. faecalis* + *E. faecium*	*E. faecalis*
90	*E. faecium*	*E. faecium*	555	*E. faecium*	*E. faecium*
92	Other enterococci	*E. durans1*	576	*E. faecalis* + *E. faecium*	*E. faecalis*
93	*E. faecalis*	*E. faecalis*	585	*E. faecalis* + *E. faecium*	*E. faecium*
94	*E. faecium*	*E. faecium*	619	*E. faecalis*	*E. faecalis*
96	*E. faecalis*	*E. faecalis*	627	*E. faecium*	*E. faecium*
99	*E. faecalis* + *E. faecium*	*E. faecium*	630	*E. faecalis* + *E. faecium*	*E. faecalis*
100	*E. faecalis* + *E. faecium*	*E. faecium*	702	*E. faecalis* + *E. faecium*	*E. faecium*
101	*E. faecalis* + *E. faecium*	*E. faecium*	704	*E. faecalis* + *E. faecium*	*E. faecalis*
102	*E. faecalis*	*E. faecalis*	714	*E. faecalis* + *E. faecium*	*E. faecalis*
106	*E. faecalis*	*E. faecalis*	718	*E. faecalis* + *E. faecium*	*E. faecium*
107	*E. faecium*	*E. faecium*	725	*E. faecalis*	*E. faecalis*
361	*E. faecalis*	*E. faecalis*	734	*E. faecalis*	*E. faecalis*
399	*E. faecalis*	*E. faecalis*			

PCR = Polymerase Chain Reaction, API = Analytical Profile Index, ID = Identity.

**Table 2 antibiotics-12-00657-t002:** Prevalence of Enterococcus in Poultry from the four Districts.

Factor	Categories	n Tested	n Positive	Prevalence (%)	95% CI
Overall	Positivity	492	153	31.1	27.1–35.4
Province	Lusaka	107	33	30.8	22.5–40.6
	Copperbelt	385	120	31.2	26.6–36.1
District	Lusaka	50	22	44.0	30.3–58.7
	Chongwe	57	11	19.3	10.5–32.3
	Kitwe	245	78	31.8	26.1–38.1
	Ndola	140	42	30.0	22.7–38.4
Enterococci isolates	*E. faecium*	153	16	10.5	6.3–16.7
	*E. faecalis*	153	58	37.9	30.3–46.1
	All other *Enterococcus* species	153	79	51.6	43.5–59.7

n = number, % = percent, CI = confidence interval.

**Table 3 antibiotics-12-00657-t003:** Prevalence of specific species across the study area.

Factors	Categories	Species	n Tested	n Positive	Prevalence (%)	95% CI
Districts	Chongwe	*E. faecium*	57	1	1.8	0.1–10.6
		*E. faecalis*	57	5	8.8	3.3–20.0
		*E. faecalis* + *E. faecium*	57	1	1.8	0.1–10.6
		*E. faecalis* + other	57	2	3.5	0.6–13.2
		Other	57	2	3.5	0.6–13.2
		Total	57	11	19.3	10.5–32.3
	Lusaka	*E. faecium*	50	5	10.0	3.7–22.6
		*E. faecalis*	50	8	16.0	7.6–29.7
		*E. faecalis* + *E. faecium*	50	4	8.0	2.6–20.1
		*E. faecium* + other	50	1	2.0	0.1–12.0
		*E. faecalis* + other	50	1	2.0	0.1–12.0
		Other	50	3	6.0	1.6–17.5
		Total	50	22	44.0	30.3–58.7
	Kitwe	*E. faecium*	245	8	3.3	1.5–6.6
		*E. faecalis*	245	26	10.6	7.2–15.3
		*E. faecalis* + *E. faecium*	245	39	15.9	11.7–21.2
		Other	245	5	2.0	0.8–5.0
		Total	245	78	31.8	26.1–38.1
	Ndola	*E. faecium*	140	2	1.4	0.2–5.6
		*E. faecalis*	140	19	13.6	8.6–20.6
		*E. faecalis* + *E. faecium*	140	8	5.7	2.7–11.3
		*E. faecalis* + *E. faecium* + other	140	1	0.7	0.0–4.5
		*E. faecalis* + other	140	2	1.4	0.2–5.6
		Other	140	10	7.1	3.7–13.1
		Total	140	42	30.0	22.7–38.4
Species	Overall	*E. faecium*	492	16	3.3	1.9–5.3
		*E. faecalis*	492	58	11.8	9.1–15.1
		*E. faecalis* + *E. faecium*	492	52	10.6	8.1–13.7
		*E. faecalis* + *E. faecium* + other	492	1	0.2	0.0–1.3
		*E. faecium* + other	492	1	0.2	0.0–1.3
		*E. faecalis* + other	492	5	1.0	0.4–2.5
		Other	492	20	4.1	2.6–6.3
		Total Isolates	492	153	31.1	27.1–35.4

n = number, % = percent, CI = confidence interval, other = unidentified enterococcus species.

**Table 4 antibiotics-12-00657-t004:** Antimicrobial Susceptibility Profiles of *Enterococcus faecalis* and *Enterococcus faecium*.

Species	Susceptibility Test Result	AMP n (%)	CHL n (%)	CIP n (%)	ERY n (%)	NIT n (%)	PEN n (%)	TET n (%)	VAN n (%)
*E. faecalis*	Resistant	37 (63.8)	35 (60.3)	44 (75.9)	54 (93.1)	37 (63.8)	31 (53.4)	58 (100)	19 (32.8)
	Susceptible	21 (36.2)	23 (39.7)	14 (24.1)	4 (6.9)	21 (36.2)	27 (46.6)	0	39 (67.2)
*E. faecium*	Resistant	14 (87.5)	6 (37.5)	13 (81.3)	16 (100)	8 (50)	8 (50)	14 (87.5)	7 (43.7)
	Susceptible	2 (12.5)	10 (62.5)	3 (18.7)	0	8 (50)	8 (50)	2 (12.5)	9 (56.3)
Total	*E. faecalis*	51 (68.9)	41 (55.4)	57 (77.0)	70 (94.6)	45 (60.8)	40 (54.1)	72 (97.3)	26 (35.1)
	*E. faecium*	23 (31.1)	33 (44.6)	17 (23.0)	4 (5.4)	29 (39.2)	34 (46.0)	2 (2.7)	48 (64.9)

n = number, % = percent, AMP = ampicillin, CHL = chloramphenicol, CIP = ciprofloxacin, ERY = erythromycin, NIT = nitrofurantoin, PEN = penicillin, TET = tetracycline, VAN = vancomycin.

**Table 5 antibiotics-12-00657-t005:** Number of Isolates Resistant against one, two, three or more Antimicrobial Classes.

Isolate (Total Number)	All Susceptible n (%)	Resistant to One Class of Antibiotic, n (%)	Resistant to Two Classes of Antibiotics, n (%)	Resistant to Three or More Classes of Antibiotics, n (%)
All *Efs* and *Efm* (74)	0 (0)	2 (2.7%)	5 (6.8%)	67 (90.5%)
*E. faecalis* (58)	0 (0)	2 (3.5%)	4 (6.9%)	52 (89.7%)
*E. faecium* (16)	0 (0)	0 (0)	1 (6.3%)	15 (93.8%)

n = number, % = percent, *Efs* = *E. faecalis*, *Efm* = *E. faecium*.

**Table 6 antibiotics-12-00657-t006:** The number of different resistance genes detected in *E. faecalis* and *E. faecium* isolates.

Resistance Gene	Total Isolates Tested	Detected				Undetected
		*E. faecalis* (58)	Proportion (*E. faecalis*)	*E. faecium* (16)	Proportion (*E. faecium*)	
*aac(6′)-Ie-aph(2″)-LA*	74	33	44.6%	12	16.2%	29
*ermA*	74	1	0.01%	1	0.01%	72
*ermB*	74	35	47.3%	10	13.5%	29
*tetK*	74	40	54.1%	8	10.8%	26
*tetM*	74	45	60.8%	10	13.5%	19
*tetL*	74	41	55.4%	8	10.8%	25
*tetX*	74	3	0.04%	2	0.03%	69
*vanA*	74	2	0.03%	0	0	72

**Table 7 antibiotics-12-00657-t007:** Resistance genes detected in *E. faecalis* and *E. faecium* isolates from poultry in Copperbelt and Lusaka Provinces.

Area	Species	#RG	Resistance Genes																TRG
			*aac*		*ermA*		*ermB*		*tetK*		*tetL*		*tetM*		*tetX*		*vanA*		
			Pos	Neg	Pos	Neg	Pos	Neg	Pos	Neg	Pos	Neg	Pos	Neg	Pos	Neg	Pos	Neg	
Lusaka Province	*E. faecalis*	13	10	3	0	15	8	5	12	1	13	0	13	0	1	12	1	12	13
	*E. faecium*	6	6	0	0	6	5	1	4	2	5	1	4	2	1	5	0	6	6
Copperbelt Province	*E. faecalis*	45	23	22	1	44	27	18	28	17	29	16	32	13	2	43	1	44	32
	*E. faecium*	10	6	4	1	9	5	5	4	6	3	7	6	4	1	9	0	10	6
Chongwe	*E. faecalis*	5	4	1	0	5	3	2	4	1	5	0	5	0	0	5	0	4	5
	*E. faecium*	1	1	0	0	1	1	0	1	0	1	0	1	0	0	1	0	1	1
Lusaka	*E. faecalis*	8	6	2	0	8	5	3	8	0	7	1	8	0	1	7	0	8	8
	*E. faecium*	5	5	0	0	5	4	1	3	2	4	1	3	2	1	4	0	5	4
Ndola	*E. faecalis*	20	13	7	0	20	12	8	11	9	15	5	16	4	0	20	0	20	16
	*E. faecium*	2	1	1	0	2	2	0	1	1	2	0	2	0	0	2	0	2	2
Kitwe	*E. faecalis*	25	10	15	1	24	15	10	17	8	14	11	16	9	2	23	1	24	17
	*E. faecium*	8	5	3	1	7	5	5	3	5	1	7	4	4	1	7	0	5	5

#RG = number of isolates in which resistance-gene detection was conducted; Pos = detected, Neg = undetected, TRG = total number of isolates containing resistance genes.

**Table 8 antibiotics-12-00657-t008:** Association between antimicrobial results and their corresponding resistance genes.

Antibiotic	Genes	*X*^2^—Value	*p*-Value
TET	*tet*	3.945	0.047 ***
ERY	*erm*	6.947	0.008 ***
VAN	*vanA*	3.795	0.051

*X*^2^ = Chi-square value; ***: *p*-Value = significant at <0.05; TET = Tetracycline; ERY = Erythromycin; VAN = Vancomycin; *tet* = all tetracycline genes (*tetM*, *tetL*, *tetK* and *tetX*); *erm* = both *ermA* and *ermB* genes.

**Table 9 antibiotics-12-00657-t009:** Primers for Enterococcus Genus and Species identification.

IDENTIFICATION PRIMERS				
Target Gene	Primer Name	Primer Sequence 5′-3′	Amplicon Size bp	References
*tuf*	*tuf-F*	TACTGACAAACCATTCATGATG	112	[65]
	*tuf-R*	AACTTCGTCACCAACGCGAAC		
*ddl*	*ddlF*	CACCTGAAGAAACAGGC	475	[66]
	*ddlR*	ATGGCTACTTCAATTTCACG		
*sodAEfm*	*sodAEfm1*	CAGCAATTGAGAAATAC	190	[67]
	*sodAEfm2*	CTTCTTTTATTTCTCCTGTA		
*sodAEfs*	*sodAEfs1*	CTGTAGAAGACCTAATTTCA	209	[67]
	*sodAEfs2*	CAGCTGTTTTGAAAGCAG		

bp = base pairs.

**Table 10 antibiotics-12-00657-t010:** Primers used for the Detection of Resistance Genes.

PRIMERS FOR RESISTANCE GENES				
Target Gene	Primer Name	Primer Sequence 5′-3′	Amplicon Size (bp)	References
aac(6′)-Ie-aph(2″)-LA	*aacF*	CAGGAATTTATCGAAAATGGTAGAAAAG	369	[69]
	*aacR*	CACAATCGACTAAAGAGTACCAATC		
*erm*A	*ermAF*	TATCTTATCGTTGAGAAGGGATT	139	[70]
	*ermAR*	CTACACTTGGCTTAGGATGAAA		
*erm*B	*ermB-1*	GAAAAGTACTCAACCAAATA	639	[71]
	*ermB-2*	AGTAACGGTACTTAAATTGTTTA		
*tet*K	*tetK-1*	TTAGGTGAAGGGTTAGGTCC	697	[72]
	*tetK-2*	GCAAACTCATTCCAGAAGCA		
*tet*M	*tetM-1*	GTTAAATAGTGTTCTTGGAG	576	[72]
	*tetM-2*	CTAAGATATGGCTCTAACAA		
*tet*L	*tetL-1*	CATTTGGTCTTATTGGATCG	456	[72]
	*tetL-2*	ATTACACTTCCGATTTCGG		
*tet*X	*tetXF*	CAATAATTGGTGGTGGACCC	468	[73]
	*tetXR*	TTCTTACCTTGGACATCCCG		
*van*A	*vanAF*	CTGCAATAGAGATAGCCGCTAACA	751	[74]
	*vanAR*	TGTATCCGTCCTCGCTCCTC		

bp = base pairs.

## Data Availability

All data supporting the reported results have been provided in this study. Any questions regarding data in this study or any supplementary data that may be required may be provided by the corresponding author upon request.

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
