# Peer review of "Determination of the Prevalence and Antimicrobial Resistance of Enterococcus faecalis and Enterococcus faecium Associated with Poultry in Four Districts in Zambia"

_antibiotics, 2023, doi:10.3390/antibiotics12040657_

Round 1

Reviewer 1 Report

In this work the Authors determined prevalence and antimicrobial resistance of E. faecalis and E. faecium strains isolated from poultry in Zambia.

The work is original, interesting and provides essential data to the existing literature. The study was carried out on representative number of samples. All figures and tables are understandable and clearly presented. The interpretation is consistent with the obtained results.

Generally the manuscript is clear and concise, but contains some little disadvantages:

-          line 339  - change “as described previously (60)” as “as described previously by Li et al. 2012)

-          line 359 – delete the word “below”

-          line 189 -222 and tables 6-8 – the genes names should be written italic

-          line 328, line 87 – why only 37 isolates were subjected and examined by API test kits?

-          Chapter 4.3.1. – add in how many samples (from 492 tested) the suspect Enterococcus colonies (isolates) were found?

-          line 34-37 -   the sentence should be deeply reorganized.  (The multidrug E. faecalis and E. faecium isolates can’t be transmitted to pathogenic bacteria. The genes can be only transmitted!)

Author Response

Please refer to response attached

Reviewer 2 Report

Please see the comments in the embedded PDF.

Author Response

I have responded to all comments you made in the attached response.

Reviewer 3 Report

This study have studied the prevalence, levels of  antimicrobial resistance and presence of antimicrobial resistant genes in Enterococcus faecalis and Enterococcus faecium isolated from poultry in the four Districts of Zambia. Overall, this study focused on a type of bacteria which is of  concern but less investigated on the surveillance on MDR part. The outcomes of this research are useful for future research studies. 

Major changes: Surprising to see bacterial names are not italics. Please change it.

In the introduction, please provide the economic impact of these bacterial pathogens and negate their importance as human pathogens. what is the impact burden on poultry industry. 

This is a well written manuscript. In methods section, manufacturer details are lacking. please mention. It can be accepted in current form. 

Author Response

I have attached a word document in which all your comments have been addressed.
